# CONTRASTIVE IMPLICIT REPRESENTATION LEARNING

## ABSTRACT

Implicit Neural Representations have emerged as an interesting alternative to traditional array representations. The challenge of performing downstream tasks directly on implicit representations has been addressed by several methods. Overcoming this challenge would open the door to the application of implicit representations to a wide range of fields. Then again, self-supervised representation learning methods, such as the several contrastive learning frameworks which have been proven powerful representation learning methods. So far, the use of self-supervised learning for implicit representations has remained unexplored, mostly because of the difficulty of producing valid augmented views of implicit representations to be used for learning contrasts. In this work, we adapt the popular SimCLR algorithm to implicit representations that consist of multiplicative filters networks and SIRENs. While methods to obtain augmentations in SIREN have been studied in the literature, we provide methods for augmenting MFNs effectively. We show how MFNs lend themselves well to geometric augmentations. To the best of our knowledge, our work is the first to demonstrate that self-supervised learning on implicit representations of images is feasible and results in good downstream task performances.

## 1 INTRODUCTION

*Implicit Neural Representations* (INRs) are functional representations of discretely sampled continuous signals. Namely, INRs are the parameters of neural network fields

$$f_\theta : \mathbb{R}^d \longrightarrow \mathbb{R}^c, \tag{1}$$

where the input dimension $d$ is the signal domain dimension and $c$ is the number of signal channels. For an RGB image, for instance, $d = 2$, $c = 3$, and its implicit representation are the parameters of a function $f_\theta$ fitted to map coordinates of a 2D grid to the corresponding RGB values of the image. Particularly, to obtain the INR of a discrete signal $\{I_i\}_{i=1}^N$ at $N$ discrete locations, we fit the parameters $\theta$ of $f_\theta$ to minimize the reconstruction loss

$$\mathcal{L}(\theta, \{I_i\}_{i=1}^N) = \sum_{i=1}^N |f_\theta(x_i) - I_i|_2^2, \tag{2}$$

where $x_i$ are the coordinate locations on the domain of the signal. Unlike discrete representations, which rely on fixed-size arrays to contain data, implicit representations are a much more natural choice for continuous signals, offering a new paradigm for representing complex, high-dimensional data in a compact, efficient manner (Park et al., 2019; Mildenhall et al., 2021; Tancik et al., 2020; Xie et al., 2022; Yin et al., 2022; Pumarola et al., 2021; Li et al., 2022). However, one of the challenges that has gained substantial attention lies in performing downstream tasks directly on these implicit representations.

### 1.1 CONTRASTIVE LEARNING

Prior to the exploration of implicit neural representations, the domain of self-supervised representation learning has shown impressive results in enabling features extraction without explicit supervision, i.e., from unlabelled data. Originally, this overcame the problem of the high cost of data annotation in supervised learning (Le-Khac et al., 2020). One of the most successful among the

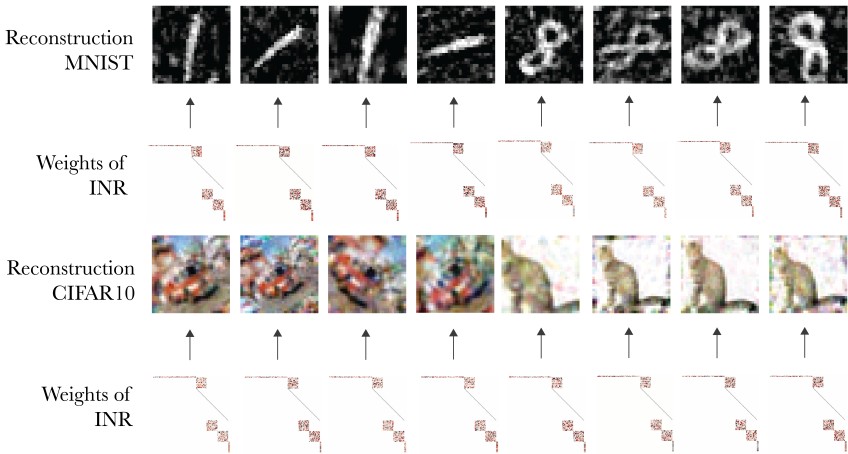

Figure 1: We train an encoder with SimCLR on *implicit neural representations* (INR). To obtain the pairs at the core of the method we augment the weights of the INR. These are standard augmentations such as random drop-out and Gaussian noise, and geometric augmentations such as rotations, translations, and scaling. The latter are constructed to obtain the desired transformation on the reconstructed image, while changing only the weights of the INR. Here we show the edge feature matrix of a *multiplicative filter network* (MFN), which also shows the connectivity of the computation graph of the neural field. This is what is ultimately used in the encoder architecture (Zhang et al., 2023). Each square block corresponds to a hidden layer in the network, the horizontal line at the top is the weights of filters, while the diagonal block shows the element-wise products used in an MFN between filters and hidden layers.

self-supervised learning frameworks is *contrastive learning* (Jaiswal et al., 2020). Despite its remarkable success in different domains and with downstream tasks, the use of self-supervised learning, and in particular contrastive learning, with implicit representations has remained unexplored. The main reason is that the operations required to perform contrastive learning on array representations of images are not straightforward to translate to their implicit representations. To the best of our knowledge, this work is the first that demonstrates the applicability of contrastive learning techniques to implicit neural representations, with the exception of Navon et al. (2023) which suggested this direction in a simplified setting with a dataset of sinusoids. We think that this is an important step forward in the representation learning domain for the following reasons:

- Neural fields are *resolution independent*. The *same* architecture can be trained to reconstruct images of different resolutions and shapes. One key challenge of self-supervised methods is learning representations that generalize well across datasets of different shapes and image sizes.

- Some modalities, such as scenes, shapes, and audio do not have array representations that can be easily adapted to work with existing self-supervised methods.

## 1.2 CONTRIBUTION

In this work, we focus on the widely acclaimed SimCLR algorithm (Chen et al., 2020). SimCLR is a contrastive learning framework for learning visual representations. At its core, it learns by maximizing the alignment of the representations of *augmented* views of the same image. The structure of SimCLR is described in Section 3. We summarize as follows the contributions of our work:

- We show how contrastive learning can be applied to implicit neural representations with different architectures.

- We characterize the permutation symmetries of multiplicative filter networks and provide further evidence for the importance of processing weights with functions that are invariant with respect to the permutation symmetries of the INRs.

## 2 DATASETS OF IMPLICIT NEURAL REPRESENTATIONS

A dataset of implicit representation is simply a set $\{\theta_j\}$ of neural field parameters, each one reconstructing an image in a dataset of images. To build a dataset of implicit neural representations several design choices have to be made, such as the network architecture and the reconstruction accuracy to name a few. Therefore, we experimented with a combination of those as reported in table 1. A crucial feature of our datasets is the number of implicit representations that have been obtained per image. For a given signal and a given neural field architecture, there exist multiple functionally equivalent implicit neural representations. This is due to the existence of multiple local minima of the optimization problem of eq. equation 2. As described in the following sections, the proposed method is aligned with the implicit working hypotheses of recent works (Navon et al., 2023; Zhang et al., 2023), namely that *permutation symmetries* of the neural fields parameterizations account for most of them. Our experiments provide further evidence for the validity of this hypothesis.

### 2.1 NEURAL FIELD ARCHITECTURES

For our expermients we use datasets of implicit representations constructed using SIRENs (Sitzmann et al., 2020) and Multiplicative Filters Networks (Fathony et al., 2020). Despite SIRENs being the most popular INR architecture in the literature, in section 3.2 we explain how multiplicative filters networks are more amenable to augmentations. We therefore present them as good candidate for contrastive learning.

*SIREN (sinusoidal representation networks) Neural Fields* (Sitzmann et al., 2020) are a specialized form of multilayer perceptron that have proven to be particularly adept at representing complex functions, including natural images and signals. They are distinguished by their sinusoidal activation functions. SIRENs utilize periodic activation functions to improve the network's capacity to capture variations in data, especially when dealing with wave-based signals and images. The structure of a SIREN network is formalized as:

$$
\begin{aligned}
h^{(1)} &= x \\
h^{(i)} &= \sin\left(\Omega_0 W^{(i-1)} h^{(i-1)} + b^{(i-1)}\right),\, i = 1, \ldots, k-1 \\
f_\theta(x) &= W^{(k)} h^{(k)} + b^{(k)},
\end{aligned}
\tag{3}
$$

where $W^{(i-1)} \in \mathbb{R}^{d_i \times d_{i-1}}$ and $b^{(i-1)} \in \mathbb{R}^{d_i}$ denote the weight matrix and bias vector for the $i$-th layer, respectively, and $\Omega_o$ is a scalar. Here, the $\sin$ function is utilized as the activation function at each layer. Their capacity to represent and reconstruct data patterns is attributed to the sinusoidal activations, enabling the network to capture a wide range of frequencies and amplitudes effectively. This feature makes SIREN particularly beneficial for tasks involving the modeling of natural signals and images.

*Multiplicative Filters Networks* (MFNs) (Fathony et al., 2020) are neural fields architectures that unlike feedforward neural networks do not rely on compositional depth for reconstruction power. Instead, MFNs apply nonlinear filters to the input and iteratively multiply together linear functions of those filters. Explicitly, an MFN is defined by the recursion

$$
\begin{aligned}
z^{(1)} &= g\left(x; \psi^{(1)}\right) \\
z^{(i+1)} &= \left(W^{(i)} z^{(i)} + b^{(i)}\right) \odot g\left(x; \psi^{(i+1)}\right),\, i = 1, \ldots, k-2 \\
f_\theta(x) &= W^{(k-1)} z^{(k-1)} + b^{(k-1)},
\end{aligned}
\tag{4}
$$

where $\odot$ represents element-wise multiplication, $W^{(i)} \in \mathbb{R}^{d_{i+1} \times d_i}$, $b^{(i)} \in \mathbb{R}^{d_{i+1}}$ and $g : \mathbb{R}^d \to \mathbb{R}^{d_i}$ are the nonlinear filters parameterized by $\psi_i$ that are applied to the input directly. Among the possible choices of the filters, throughout the paper, we use a linear layer composed with a sine function, $g\left(x; \psi^{(i+1)}\right) = \sin \omega x + \phi$ where $\omega_i \in \mathbb{R}^{d \times d_1}$ and $\phi_i \in \mathbb{R}^{d_1}$. In their paper, Fathony et al. (2020) prove that such a multiplicative filter network is ultimately just a linear function of an exponential (in $k$) number of Fourier basis functions.

## 2.2 PERMUTATIONS SYMMETRIES

It has been well known (Hecht-Nielsen, 1990) that the parameter space of neural networks is characterized by a combinatorial number of permutation symmetries. In particular, consider any layer of a MLP $W^{(i+1)}\sigma(W^{(i)}z^{(i)} + b^{(i)})$ and permute the weight matrices and the bias vector as $W^{(i)}, b^{(i)} \mapsto P^T W^{(i)}, P^T b^{(i)}$, and $W^{(i+1)} \mapsto W^{(i+1)} P$. The result is a *different* implicit representation that nonetheless represents the *exact* same function. In the literature, permutation symmetries of neural networks have been studied, mostly to investigate the loss landscape of neural networks (Chen et al., 1993; Ainsworth et al., 2022; Simsek et al., 2021; Entezari et al., 2021).

Similarly to MLPs, we can characterize the permutation symmetries of MFNs. It is easy to see from eq. 4 that for a MFN parameterized by $W^{(i)}$, $b^{(i)}$, $\omega^{(i)}$, and $\phi^{(i)}$, any set of $k-1$ permutations $(P_1, \ldots, P_{k-1})$ acting of the weight space as:

$$
\begin{aligned}
W^{(i)} &\mapsto P_{i+1}W^{(i)}P_i^T; \quad b^{(i)} \mapsto P_{i+1}b^{(i)} \quad 1 \le i \le k-2 \\
\omega^{(i)} &\mapsto P_i\omega^{(i)}; \quad \phi^{(i)} \mapsto P_i\phi^{(i)} \quad 1 \le i \le k-1 \\
W^{(k-1)} &\mapsto W^{(k-1)}P_{k-1}^T; \quad b^{(k-1)} \mapsto b^{(k-1)}
\end{aligned}
\tag{5}
$$

defines a symmetry.

In light of recent findings in the literature, and the results of the experiments presented in this paper, we make the educated hypothesis that *permutation symmetries are responsible for a good amount of the impracticability of downstream tasks*. This stems from the observation that, once permutation symmetries become irrelevant due to the permutation invariance of the encoder, the latter can easily align (high top-k validation accuracy) vector representations of differently-initialized INRs. It is essential to note that this hypothesis is speculative and not a formal statement, serving as a foundation for further investigation and discussions rather than a conclusive assertion. It is informed by current insights and aims to stimulate further research and exploration into this intricate area.

## 3 METHOD

In this section, we outline the details of the method. We start with a brief description of the SimCLR framework. It should be noted that the overview we provide is a concise summary and not exhaustive. For a comprehensive understanding and detailed insights into the SimCLR algorithm, readers are encouraged to refer to the original paper (Chen et al., 2020).

### 3.1 SIMCLR

SimCLR (Contrastive Learning of Visual Representations) is a self-supervised learning algorithm introduced for the efficient learning of visual representations. It operates by maximizing the similarity between augmented views of the same data instance while minimizing the similarity between augmented views of different instances. In particular, the SimCLR architecture consists of an *encoder* $f(\cdot)$ and a small MLP *projector head* $g(\cdot)$. SimCLR starts by randomly sampling a minibatch of $N$ examples and generating two distinct augmented views (*positive pairs*) $\tilde{x}_i$ and $\tilde{x}_j$ for every example. All the augmented views in the minibatch are passed through the encoder and the projector to get $\mathbf{z}_i$ and $\mathbf{z}_j$. The objective of SimCLR is defined by the *contrastive loss function*, typically the Noise Contrastive Estimation (NCE) loss or the Normalized Temperature-Scaled Cross Entropy Loss (NT-Xent). It is formulated, for a positive pair of examples $(i, j)$ as:

$$
\ell(i,j) = -\log\left(\frac{\exp(\text{sim}(\mathbf{z}_i, \mathbf{z}_j)/\tau)}{\sum_{k=1}^{2N} \mathbb{I}_{i \neq k} \exp(\text{sim}(\mathbf{z}_i, \mathbf{z}_k)/\tau)}\right),
\tag{6}
$$

where

$$
\text{sim}(\mathbf{z}_i, \mathbf{z}_j) = \frac{\mathbf{z}_i \cdot \mathbf{z}_j}{\|\mathbf{z}_i\|\|\mathbf{z}_j\|}
$$

is the cosine similarity between vectors $z_i$ and $z_j$, $\mathbb{I}_{i \neq k} \in \{0, 1\}$ is the indicator function, and $\tau$ is a temperature parameter that scales the similarities. Intuitively, the contrastive learning task aims to

identify $\tilde{x}_j$ in $\{\tilde{x}_k\}_{k \neq i}$ for a given $\tilde{x}_i$. Once the contrastive objective has been optimized, the head projector $g(\cdot)$ is thrown away and the encoder $f(\cdot)$ is used to get the representations to be used for downstream tasks.

## 3.2 DATA AUGMENTATIONS

As seen, SimCLR and other contrastive self-supervised learning methods such as MoCo (Momentum Contrast for Unsupervised Visual Representation Learning) (He et al., 2020), and BYOL (Bootstrap Your Own Latent) (Grill et al., 2020), relies on augmentations to systematically define the contrastive prediction task. For discrete pixel representations of images, common augmentations can be broadly categorized into two types based on the nature of the transformation applied to the image. The first category encompasses spatial or geometric transformations, which involve altering the structural form of the data. Examples of these transformations include cropping and resizing often accompanied by horizontal flipping, and rotation, as noted by (Gidaris et al., 2018), and cutout (DeVries & Taylor, 2017). The second category is characterized by appearance transformations that primarily focus on altering the visual aesthetics of the image without changing its structural integrity. Such augmentations include color distortions like color dropping, and adjustments to brightness, contrast, saturation, and hue, as explored by Howard (2013) and (Szegedy et al., 2015). Additionally, other transformations like Gaussian blur and Sobel filtering fall under this category of augmentations.

Particularly when employed for the type of tasks considered in this work, namely classification, augmentations are transformations of datapoints that preserve their *object identity*. It is not straightforward to perform any systematic transformation on implicit neural representations in such a way that the *object identity* of the image they represent is preserved. In other words, it is easy to destroy any semantic information contained in a neural field by acting on its parameters. Here we show how augmentations can be performed on implicit representations to enable contrastive learning. We divide the augmentations into three categories: *standard*, *geometric*, and *random seed* augmentations.

**Standard augmentations** With standard augmentations we refer to those transformations performed on the datapoints that are commonly used in machine learning to randomly alter the dataset and add some regularization effect to the training. As in Navon et al. (2023), in this work, we use Gaussian noise and random drop-out.

**Geometric augmentations** With geometric augmentations we refer to the action of certain groups of transformations on the functions that the implicit representations define (Navon et al., 2023). Formally, let $G$ be a group of transformations such as the group of rotations or the translation group, and let $f_\theta : \mathbb{R}^2 \to \mathbb{R}^3$ be a neural field representing an image, as standard practice, we define the group action of $g \in G$ on the set of functions as

$$L_g f_\theta(x) = [f \circ g^{-1}](x) = f(g^{-1}x) \tag{7}$$

Operationally, this means that the value of the $g$-transformed function $L_g f_\theta(x)$ at the point $x$, is the value of the original function $f$ at the point $g^{-1}x$, which is the unique point mapped to $x$ by $g$. For example. At this point, to define the augmentation $t_g : \theta \mapsto t(\theta)$, we need to find a transformation of the weights $\theta$ such that

$$f_{t_g(\theta)}(x) = L_g f_\theta(x) = [f \circ g^{-1}](x) = f(g^{-1}x) \quad \forall x \in \mathbb{R}^2. \tag{8}$$

For transformations such as rotations and scaling, their group action on $\mathbb{R}^2$ is simply a matrix multiplication, i.e., for every $g$, $g^{-1}x = R_g x$ for some $R \in \mathbb{R}^{2 \times 2}$. It is straightforward to note that, for MLP, the action of $t_g$ on $\theta$ simply consists of multiplying from the right by $R_g$ the first weight matrix. In the case of MFNs, it consists of multiplying from the right by $R_g$ for every filter matrix. For translations, $t_g$ does not affect weight matrices but acts on the biases of the first layer in the case of MLPs as $t_g(b^{(1)}) = b^{(1)} - W^{(1)}t$, and all the biases in the filter layers in the case of MFNs as $t_g(\phi^{(i)}) = \phi^{(i)} - \omega^{(i)}t, \quad i - 1, \ldots, k - 1$.

At this point, it is worth noting that for different architectures, a different proportion of parameters is affected by augmentations. In general, for contrastive learning to extract the relevant features from a dataset of INRs, the more these are affected by augmentations, the better. For MFNs, the proportion of weights affected by geometric augmentations is considerably higher than MLPs. For

example, geometric augmentations on MLPs, alter is $\frac{3}{d \times (n-1)+3}$, whereas in an MFN with $n$ layers and hidden dimension $d$ the proportion of weights altered by geometric augmentations is $\frac{1}{(d+1)+\frac{1}{n}}$. For example, for an MFN with 4 layers and hidden dimension 4, this proportion is about 0.19, which is considerably higher than the proportion of 0.09 with a 2 hidden layer MLP with hidden dimension 32.

**Random seed augmentations** Fitting a single image from different initializations results in different INRs. Our datasets are therefore made of multiple INRs for every image, obtained from different initializations. During training, two positive pairs are always obtained augmenting *different* INRs obtained starting from different initializations.

### 3.2.1 ALIASING

It can happen that two completely different INRs are indistinguishable when sampled at discrete locations, because of aliasing. A basic result in signal processing is given by the Nyquist-Shannon sampling theorem. This states that to sample a finite band signal without loss of information, it must be sampled with a frequency at least double the frequency of the spectral component of the information signal at a higher frequency (also called Nyquist frequency). It is easy to compute the highest frequency of an MFN: it is the sum of the maximum frequencies of filters, which ultimately are the absolute values of entries of the filter matrices. We therefore propose to add the following regularizer to the reconstruction loss equation 2:

$$\mathcal{R}(\theta) = \sum_{i=1}^{k-1} |\omega_i|_0, \tag{9}$$

where $|\cdot|_0$ is the $L_0$ norm.

As reported in Fathony et al. (2020) initializations are crucial to get good reconstruction accuracy. Empirically, we find that initializations are also key to avoiding aliasing. Our regularizer obviates the need to find a trade-off between good reconstruction and aliasing, by allowing us to initialize the MFN with higher frequencies around the Nyquist, and keeping them below while fitting. Figure 2 shows the effects of the proposed regularization.

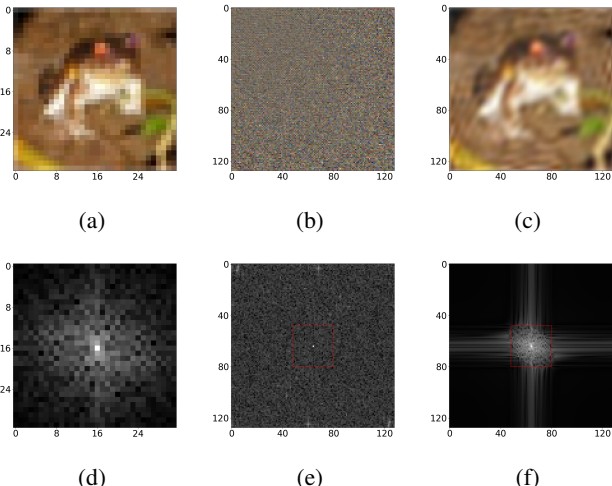

Figure 2: Aliasing in multiplicative filters networks. (a) An MFN trained without anti-aliasing regularization sampled on the training grid. (b) The *same* MFN sampled on a finer grid. (c) An MFN trained with anti-aliasing regularization. (d) The frequency spectrum of the original image. (e) The frequency spectrum computed with a higher spacial resolution. Outside the box frequencies shold be zero. (f) The frequency spectrum of an MFN trained with anti-aliasing regularization.

### 3.3 ENCODER NETWORK ARCHITECTURE

Our encoder network is based on the work of Zhang et al. (2023). In this work, the authors propose to use the computational graph of neural networks and encode INRs with graph networks or transformers that respect the permutation symmetries present in the parameter space. Under the computational graph paradigm, the biases of each layer correspond to node features, while the weights of each layer correspond to edge features. For a standard fully-connected MLP, the edge features matrix is organized as a block-superdiagonal matrix, i.e. a block matrix with blocks populated 1 above and to the right of the main diagonal (see Figure 1). The authors extend recent graph network and transformer architectures, namely PNA (Corso et al., 2020), and Relational Transformer (Diao & Loynd, 2023) to better accommodate edge features since the bulk of the information in the computational graph is in the edge features.

## 4 EXPERIMENTS

We evaluate our method on CIFAR10 and MNIST. We compare it to a supervised learning method that uses the same architecture that we use as the encoder in SimCLR. Our code and datasets will be made publicly available upon acceptance.

### 4.1 DATASET

| **SIREN** | | | **MFN** | | |
|---|---|---|---|---|---|
| Parameter | MNIST | CIFAR10 | Parameter | MNIST | CIFAR10 |
| layers | | 3 | filters | | 3 |
| hidden_dim | | 32 | hidden_dim | | 16 |
| omega_0 | | 30 | input_scale | 3.03 | 3.46 |
| learn_rate | | $5 \times 10^{-4}$ | learn_rate | | $10^{-2}$ |

Table 1: Hyperparameters used in the datasets of Implicit Neural Representations.

We train our model using four INR datasets. Of these, two of them are obtained by fitting on MNIST, while the other two by fitting on CIFAR10. For both datasets, 30 INRs are trained *per* image and used for random seed augmentations. For details on the hyperparameters used to obtain the datasets, see Table 1.

### 4.2 EXPERIMENTAL SETUP

We use the Relational Transformer architecture from Zhang et al. (2023) *without probe features* for both contrastive learning and supervised learning experiments. Essentially, probe features are the activations of every layer, *including* the output layer, obtained using learnable inputs. We chose not to use those in our experiments, to show that our method lears *in weight space* and does not require querying the neural field to perform well. The architecture is the same for all experiments. The optimizer is Adam (Kingma & Ba, 2014) with different learning rates for each experiment. We noticed that the learning rate had a great impact on the ability of the model to fit the data.

When performing augmentations for contrastive learning we first load a batch of two random seeds of INRs fit to the same images. Then, we apply random augmentations from the set described in the previous section to each INR.

### 4.3 RESULTS

We first looked at the embeddings obtained using the learned encoders and compared them to the weights of the INRs using t-SNE Van der Maaten & Hinton (2008). The contrastive method successfully results in structured embeddings in MNIST, as seen in Figure 3. We found the model struggling with CIFAR10 for both SIREN and MFN, because of the increased complexity of the data.

|            | SIREN |         | MFN   |         |
| ---------- | ----- | ------- | ----- | ------- |
| Dataset    | MNIST | CIFAR10 | MNIST | CIFAR10 |
| *Supervised*  | 95.16 | 59.28   | 54.71 | 40.21   |
| *Contrastive* | 53.01 | 36.01   | 42.97 | 30.71   |

Table 2: Accuracy (%) for all INRs used, with both the supervised method and a linear probe on the features obtained using the frozen encoder. We used the best accuracy across different runs for all models.



Figure 3: Embeddings of the datasets using t-SNE. On the left, we used the weights of the INRs without any processing, no structure is present and they are distributed seemingly at random. Follow the MFN CIFAR10 and SIREN CIFAR10 (in this order). This dataset is harder to fit, however, some structure emerges. Finally, MFN MNIST and SIREN MNIST. Different colors represent different class labels.

We then measured the accuracy of the supervised method on the validation set. For the contrastive methods, we fit a linear probe on the embeddings predicted using the frozen encoder on the training set. The contrastive method surprisingly performs more closely to the supervised one on the MFN, compared to SIREN. This suggests that performing contrastive learning is more beneficial for certain architectures of neural fields. Further work is needed to investigate what architectures self-supervised learning performs better.

## 5 RELATED WORK

**Implicit Neural Representations** The design of deep learning architectures to process the parameters of neural networks is a relatively new research direction. Here we provide an overview of the most relevant pioneering studies in this field.

The works of Eilertsen et al. (2020) and Unterthiner et al. (2020) are centered around predicting attributes of trained neural networks (NNs) by examining their weights. Eilertsen et al. (2020) focuses on estimating the hyperparameters employed during the network's training phase, while Unterthiner et al. (2020) is dedicated to assessing the network's capacity for generalization. Both investigations involve the application of standard NNs to the flattened weights or their statistics. Xu et al. (2022) introduced a concept wherein NNs are processed through the application of another NN to a combination of their high-order spatial derivatives, a technique particularly suited for implicit neural representations (INRs) where derivative information is pertinent. However, the adaptability of these networks to broader tasks remains ambiguous, and the necessity for high-order derivatives can impose a significant computational load. Dupont et al. (2022) proposed a novel approach to perform deep learning tasks like generative modeling, on a collection of INRs. They advocated for the meta-learning of small vectors, referred to as modulations, which are integrated into a neural network, with parameters shared across all training instances, to achieve meaningful data representations. In our work, we opted not to use conditioned neural fields nor the meta-learning initialization technique such as the one proposed by Tancik et al. (2021). This is to test how our method performs with out-of-the-shelf implicit representations that can be obtained easily, without the need of the shared-across-networks parameters nor the meta-learned initialization that might not work as a good initialization for different datasets. Finally, relevant works for our method are certainly that of Zhang et al. (2023), from which we adapted the proposed transformer-like architecture to work with multiplicative filters networks, and that of Navon et al. (2023); Zhou et al. (2023), which first

demonstrated the importance of augmentations and permutation invariant architectures for processing weights of neural fields.

**Contrastive Learning** The field of Self-Supervised Learning (SSL) is rapidly advancing, focusing on utilizing unlabeled visual data. Contemporary strategies primarily depend on comparing embeddings derived from transformed input images. This approach is rooted in the concept of aligning image representations subjected to minor alterations, a notion introduced by Becker and Hinton (Becker & Hinton, 1992). In this context, SSL techniques fall into two primary classifications: contrastive learning and non-contrastive learning. This study narrows its exploration to contrastive learning methods such as MoCo (He et al., 2020) and BYOL Grill et al. (2020) and in particular to SimCLR Chen et al. (2020). Relevant to our study is the work of Schürholt et al. (2021) where they propose to perform self-supervised learning on the weights of neural networks to predict model characteristics. They differ from us as they do consider INRs and do not use an encoder that is invariant to permutations. Therefore propose to use permutations as augmentations. Finally, Navon et al. (2023) tested their permutation invariant architecture in a simplified contrastive learning setting.

## 6 CONCLUSION

In conclusion, this research has demonstrated the applicability and extensive potential of self-supervised learning to implicit neural representations. Our findings spotlight SSL as an interesting research direction in the field of implicit representations, showcasing its ability to effectively learn useful representations from unlabeled datasets of INRs. In that regard, one key finding is the importance of the random seed augmentations, as described in section 3.2.

We propose MFNs as a candidate INR architecture for the larger proportion of parameters that are affected by geometric augmentations, such as rotations, scaling, and translations. We also provide a method to regularize MFNs and avoid aliasing. Other than obtaining good reconstructions, this method also results in a constrained implicit representation space. We find that fitting regularized INRs results in better downstream performances.

Last, our experiments provide further evidence for the hypothesis that permutation symmetries represent the most significant challenge in processing the weights of neural networks. This stems from the observation that very expressive architectures fail to align, in terms of top-1 and top-5 validation accuracy, positive pairs obtained with random seed augmentations. Conversely, permutation invariant architectures rapidly achieve good alignment.

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
