# OpenReview forum: "Contrastive Implicit Representation Learning"
_ICLR.cc/2024/Conference — ICLR 2024 Conference Withdrawn Submission_

### Official Review · Reviewer_QeAy · 2023-10-23

**Soundness:** 2 fair
**Presentation:** 2 fair
**Contribution:** 1 poor
**Rating:** 1
**Confidence:** 3

**Summary:**

This paper introduces Contrastive Implicit Representation Learning, which is an application of SimCLR to implicit representations. Implicit representations are functional representations of discretely sampled continuous signals. They compare supervised and unsupervised learning for such representations. For unsupervised learning, they use the SimCLR algorithm.

**Strengths:**

- I have not come across much work (if any) that studies implicit representations. Thus, I find it interesting to see something in this direction.
- The math in Section 2 and 3 is mostly correct.

**Weaknesses:**

**Major**

- **Table 2**: I’d like to see the average performance and standard deviations (in brackets next to the average performance) rather than the best accuracy for all runs. Reporting best accuracy is cherry-picking and not representative. Please change this accordingly. Why do you think the performance for contrastive learning is so poor compared to supervised learning? Do you have an explanation or at least an idea? The MNIST results look pretty bad.


- **Section 4.2**: Why do you think the learning rate has a large impact on the model’s data fitting ability? This seems to be brittle.


- **Figure 3**: I don’t see any useful structure emerging from t-SNE visualizations for CIFAR-10. The visualization indicate that the representations are almost entirely entangled. For MNIST it looks fine, but I think we are past MNIST in Computer Vision.


- **Section 4.3**: You write “*The contrastive method surprisingly performs more closely to the supervised one on the MFN, compared to SIREN. This suggests that performing contrastive learning is more beneficial for certain architectures of neural fields.*” I am not convinced that you can draw the latter conclusion. I am not surprised at all by this finding. The performance of SIREN is pretty strong for the supervised learning method. This is close to what I would expect for a good classifier to achieve on MNIST (CIFAR-10 is a bit weaker than what I would expect for a good classifier but it is not terrible). However, the performance of MFN using supervised learning is pretty poor. Contrastive seems to work poorly irrespective of whether you use SIREN or MFN, although it is even worse for MFN (honestly, the Contrastive + MFN results are one of the worst performances I have ever seen for MNIST and CIFAR-10). So, it is pretty obvious that contrastive and supervised learning performances are closer for MFN than they are for SIREN. Neither MFN nor contrastive seem to yield useful representations. So, it is expected that for MFN there's not a big difference between supervised and contrastive. The only useful representations that seem to emerge are the ones found by using supervised learning in combination with SIREN.


- Findings re augmentations: It is not great that most variance is attributed to the random seed “augmentations” and that they affect performance more than any of the other augmentations. Why do you think the initialization matters so much here?


**Minor**

- Why is the signal domain dimension $d = 2$? Is this because of height and width and each of the two defines a single dimension? This is not clearly defined anywhere. Although this might be trivial to you, a reader can only guess why the signal domain dimension $d = 2$. This is not great. There should not be a need to guess.


- In the Introduction you define a discrete signal as ${\{I_{i}\}}_{i=1}^{N}$. What is $I$? This is not defined anywhere. Again, I can only guess that you refer to an image by $I$. However, it may not be obvious to everyone. People use different notations for inputs. I usually use $x$ rather than $I$, even if the input is an image.


- There is a typo in the second line of subsection 1.1. It should read “feature extraction” and not “features extraction”.
In reason one of subsection 1.1, you claim that “[...]. One key challenge of self-supervised methods is learning representations that generalize well across datasets of different shapes and image sizes.” I get the latter point. Different datasets have different image sizes. But what do you mean by differently shaped datasets and how is the “poor” generalization performance of contrastive learning across “differently shaped datasets” quantified? I’d be curious to better understand this.


- **Section 2**: “a dataset of implicit representation (*NOTE: this should be plural and not singular I suppose*) is simply a set $\{\theta_{j}\}$. $j$ seems to be an index. From where to where is this index running?
First paragraph in section 2: Another typo. What is “eq. equation 2”? Should probably read “Eq. 2”, “eq. (2)”, or “Equation 2.”.


- **Section 3.2**: What do you mean by *object identity*? Could you elaborate?

**Questions:**

See weaknesses. Although I find the idea interesting, I am not convinced by the results. This work needs another (major) iteration and more experiments.

---

### Official Review · Reviewer_mwVc · 2023-10-24

**Soundness:** 3 good
**Presentation:** 3 good
**Contribution:** 1 poor
**Rating:** 3
**Confidence:** 3

**Summary:**

This work attempts to apply contrastive learning to implicit representations (that is, functional representations) of data. There are two major obstacles: (a) Different functionals can represent the same data, and when parameterized by a neural network, different network weights can yield the same functional; (b) It is not very clear how to perform semantic-preserving data augmentation upon implicit representations.

For (a), the solutions in this work are: (i) Random seed augmentation, which fits the same data with neural networks initialized from different random seeds, in order to obtain the equivalence class of implicit neural representations; (ii) Regularizer in Eqn. (9); (iii) Use a graph neural network that respects permutation symmetries.

For (b), this work uses standard augmentations including Gaussian noise and random drop-out, as well as geometric augmentations, mainly rotation and translation. This work does not use what the authors call “appearance transformations”, such as color distortion which is quite important for SimCLR to achieve high performance.

**Strengths:**

The subject matter of this work is very interesting. I also believe that representation learning (such as contrastive learning) for implicit representations of data is a very important problem, especially for Nerfs. For this problem, the authors point out the main challenges, and propose some solutions such as geometric and random seed augmentations, which are quite reasonable. Overall, the manuscript is quite readable, and does provide some food for thought.

**Weaknesses:**

The major weakness of this work is that it seems quite half-baked and shallow, and its contributions might be insufficient for a conference publication.

Regarding the proposed methods, the new ones that, as far as I know, did not appear in prior work are the geometric augmentations for INR, random seed augmentations, and the regularizer Eqn. (9). These ideas, while they do make sense, seem more like some initial thoughts rather than well-formulated methods. And the justification for the proposed methods are quite weak. Theoretically, there is no guarantee that these methods would work. Empirically, in Table 2, the performance of contrastive learning seems much worse than supervised learning.

Of course, a paper could be a good paper if it raises an important question that arouses the awareness of the community, even without providing a solution. But I do not think this work makes sufficient contributions in this direction either. First of all, learning representations for implicit representations is not a new problem. There is lots of work on this in the Nerf community. Second, I don’t think the authors catch the real central question. From my understanding, the central question is this: If images are represented in a format other than pixel values (that is, Euclidean), can one still perform contrastive learning? In fact, there are lots of representations of images, including Fourier representation, wavelet representation, and the neural representation studied in this work. The main challenge is can we find semantic-preserving data augmentations for each one of these representations. For example, for Fourier representation, it could be the case that high frequency components have little effect on the semantics, so altering these components could be one such augmentation. After reading this submission, I get the impression that contrastive learning is not very feasible for INR, mostly because the space of INR does not have a basis like the Fourier basis, so the aliasing problem seems very hard to solve.

Above I provided some suggested directions and possible ideas to the authors. There are of course other directions the authors could explore, such as the effect of the network architecture on the structure of the INR space. Also, it is totally fine if the authors choose to only focus on INR, but in that case I think the authors need to carry out a much more extensive theoretical analysis and/or empirical study.

**Questions:**

Overall, I think the problem studied in this work is really interesting, and I encourage the authors to explore further in this direction. However, the current manuscript does not make enough contributions. The ideas are not well formulated, there is little theoretical analysis, and the empirical evidence is weak. Thus, unfortunately I have to recommend rejection for this submission.

With that said, I think this manuscript could be an early draft of a really great paper, if the authors do one or more of the following:
- Study more representations such as Fourier representation, wavelet representation, etc. Investigate whether contrastive learning is feasible for these representations, and if so, whether it provides any benefits over the standard pixel value representation.
- Currently, Nerf is arguably the most popular implicit representation, so I think the authors need to compare their work to representation learning for Nerfs.
- Make a more in-depth theoretical analysis of INR. For example, can you find a basis for the space of INR. If such a basis exists, then every INR can be uniquely represented by a linear combination of the basis, which immediately solves the aliasing problem.

---

### Official Review · Reviewer_1Uny · 2023-11-01

**Soundness:** 2 fair
**Presentation:** 3 good
**Contribution:** 2 fair
**Rating:** 3
**Confidence:** 3

**Summary:**

The paper shows that contrastive, self-supervised methods can be used to learn implicit neural representations. They provide methods for augmenting multiplicative filter networks which can be used for contrastive learning. They perform experiments with SimCLR on MNIST and CIFAR10 and show that the implicit representations learned using this method results in good downstream performance.

**Strengths:**

- Overall the paper is well written. The motivation and contributions are clear.

- The ideas are interesting and using self-supervised methods for implicit neural representations is not well explored.

- The experimental setup is reasonable, and the experimental insights could be useful to others such as that permutation symmetries provide a significant obstacle for training INR's .

**Weaknesses:**

- The experiments are only on CIFAR10 and MNIST which are small-scale. It's unclear if the proposed augmentations generalize to more realistic and impactful datasets such as ImageNet. Given that the goal of self-supervised methods are to scale to large datasets (LAION, JFT, Instagram-2B, etc.) which are weakly or not labeled, I think it's reasonable to show results on at least ImageNet.

- The figures aren't particularly informative. For those unfamiliar with MFN's, figure 1 is difficult to interpret. A figure illustrating the main idea of the paper would be useful. Also the table of hyper-parameters (Table 1) does not convey much information. It may be better in the appendix.


Minor Comments:

- It would be more readable if the SIREN and MFN acronym in the abstract are introduced before using the term.
- The table of hyper-parameters (Table 1) is not that informative, especially since it's the first table of the paper. It may be better in the appendix.

**Questions:**

- How computationally expensive is the method to train? Can this be scaled to reasonably sized datasets?

---

### Author Response · Authors · 2023-11-23

We would like to thank the reviewers for their time and constructive criticism. We will make use of the valuable comments to improve the quality of this work.